# A deep learning algorithm to translate and classify cardiac electrophysiology

**Parya Aghasafari[1], Pei-Chi Yang[1], Divya C Kernik[2], Kazuho Sakamoto[3], Yasunari Kanda[4], Junko Kurokawa[3], Igor Vorobyov[1,5], Colleen E Clancy[1]\***

[1]Department of Physiology and Membrane Biology, University of California, Davis, Davis, United States; [2]Washington University in St. Louis, St. Louis, United States; [3]Department of Bio-Informational Pharmacology, School of Pharmaceutical Sciences, University of Shizuoka, Shizuoka, Japan; [4]Division of Pharmacology, National Institute of Health Sciences, Kanagawa, Japan; [5]Department of Pharmacology, University of California, Davis, Davis, United States

**Abstract** The development of induced pluripotent stem cell-derived cardiomyocytes (iPSC-CMs) has been a critical in vitro advance in the study of patient-specific physiology, pathophysiology, and pharmacology. We designed a new deep learning multitask network approach intended to address the low throughput, high variability, and immature phenotype of the iPSC-CM platform. The rationale for combining translation and classification tasks is because the most likely application of the deep learning technology we describe here is to translate iPSC-CMs following application of a perturbation. The deep learning network was trained using simulated action potential (AP) data and applied to classify cells into the drug-free and drugged categories and to predict the impact of electrophysiological perturbation across the continuum of aging from the immature iPSC-CMs to the adult ventricular myocytes. The phase of the AP extremely sensitive to perturbation due to a steep rise of the membrane resistance was found to contain the key information required for successful network multitasking. We also demonstrated successful translation of both experimental and simulated iPSC-CM AP data validating our network by prediction of experimental drug-induced effects on adult cardiomyocyte APs by the latter.

*For correspondence:
ceclancy@ucdavis.edu

## Introduction

The development of novel technologies has resulted in new ways to study cardiac function and rhythm disorders (*Shaheen et al., 2018*). One such technology is the induced pluripotent stem cell-derived cardiomyocyte (iPSC-CMs) in vitro model system (*Leyton-Mange et al., 2014*). The iPSC-CM system constitutes a powerful in vitro tool for preclinical assessment of cardiac electrophysiological impact and drug safety liabilities in a human physiological context (*Sun et al., 2012*; *Lan et al., 2013*; *Burridge et al., 2016*; *Doss and Sachinidis, 2019*; *Collins et al., 2020*; *Wu et al., 2019*). Moreover, because iPSC-CMs can be cultured from patient-specific cells, it has shown to be an ideal model system for patient-based medicine (*Wu et al., 2019*; *Sayed et al., 2016*; *Matsa et al., 2016*).

While utilization of in vitro iPSC-CMs allows for testing of responses to drugs and understanding physiological mechanisms (*Tveito et al., 2018*; *Tveito et al., 2020*; *Sube and Ertel, 2017*; *Navarrete et al., 2013*), there is still a major inherent limitation of the approach: the complex differentiation process to create iPSC-CMs results in a model of cardiac electrical behavior that resembles fetal cardiomyocytes. Hallmarks of the immature phenotype include spontaneous beating, immature calcium handling, presence of developmental currents, and significant differences in the relative contributions of repolarizing potassium currents compared to adult cardiomyocytes (adult-CMs) (*Lieu et al., 2013*; *Veerman et al., 2015*; *Tu et al., 2018*). The profound differences between the immature iPSC-CMs and the adult-CMs have led to persistent questions about the utility and

applicability of the iPSC-CM action potential (AP) to predict relevant drug impacts on adult human electrophysiology (*Blinova et al., 2018*; *Sala et al., 2017*).

Several recent studies have proposed computational frameworks to address the primary limitation in using iPSC-CMs and animal cardiomyocytes for drug screening (*Tveito et al., 2018*; *Tveito et al., 2020*; *Gong and Sobie, 2018*; *de Korte et al., 2020*). The innovative studies described by Tvieto and colleagues (*Sayed et al., 2016*; *Matsa et al., 2016*) presented a translation algorithm that identified a mapping function to identify the relationships between the parameters that are defined by key ion channel conductances in the iPSC-CM APs and the adult-CM APs. In another study by Gong and Sobie, additional insights were revealed through application of an efficient partial least squares regression (PLSR) methodology to translate key physiological features between iPSC-CMs and adult-CMs. They also demonstrated the potential to translate between species, between drug-free and simple drugged models, as well as between healthy and diseased phenotypes (*Gong and Sobie, 2018*). Koivumäki et al. also tried to address the problem of iPSC-CMs immaturity by establishing a novel in silico mathematical model for iPSC-CMs, which can estimate adult-CM behavior (*Koivumäki et al., 2018*).

The efficacy of the linear translation algorithms used in the earlier studies relies on a collection of underlying assumptions (*Gong and Sobie, 2018*). One described by Tvieto et al. is that cardiac protein expression levels would differ but their functional properties remain invariant during maturation and that a drug will modify protein function in the same way for iPSC-CMs and the adult-CMs (*Tveito et al., 2018*). Tveito et al. also acknowledged the difficulty in minimizing the cost function that measures the differences between the initial and target parameters, which therefore required a brute force search algorithm for minimization. One possible explanation for the difficulty in cost function minimization is that linear translation may not capture the nonlinearities comprising the actual underlying physiological differences (*Gong and Sobie, 2018*). Another underlying assumption with linear translation is the required representation of drug effects as a simple pore block, modeled as a reduction in the maximal conductance of the channel (*Tveito et al., 2018*; *Gong and Sobie, 2018*). The earlier studies employed a biased method in that they rely on a priori parameter identification and extraction from voltage and calcium traces to allow feature mapping from immature to mature conditions (*Tveito et al., 2018*; *Gong and Sobie, 2018*). Earlier translators must also consider drug-free and drugged conditions independently.

In this study, we describe a deep learning multitask network that simultaneously performs translation and classification of signals from simulated cardiac myocytes for both drug-free and drugged conditions and demonstrate its utility for translating and predicting experimental data as well. The multitask network is an unbiased approach in that the user does not predefine the important parameters of the system. Rather, the network learns from the data to define important parameter regimes and data ranges. The new approach is indifferent to the underlying form of the models and can translate time-series data from any source. Moreover, the deep learning approach accepts nonlinearity of the system, makes no assumptions about changes in cardiac protein expression and function during maturation, and can successfully translate simple pore block and complex conformation state-dependent channel–drug interaction. The network learns from all of these data sources suggesting its broad applicability, but it requires multiple quality datasets for robust and successful translation. In addition, the multitask behavior of the network presents a single process that can perform translating any cardiac AP into the subject of perturbation.

There are multiple reasons why cell classification was considered in the study. Importantly, iPSC-CMs are generally used to understand how perturbation to the cells will result in a change to cardiac electrophysiology. Genetic and drug-induced perturbations have been commonly studied using iPSC-CM lines. An important aspect iPSC-CMs is the inherent variability reported in measurements. Indeed, wide-ranging behavior has been reported to spontaneously occur even from cells cultured from the same genetic line. Thus, it can be difficult at times to determine if a perturbation indeed has an effect compared to a control cell. Therefore, one purpose of the classification task as described in this study is to allow sorting of cells into categories without perturbation and cells that have undergone perturbation. The classification task allows us to then address the question of whether translation is effective in the setting of perturbation. We demonstrate here the efficacy of a deep learning network to perform classification in the example setting of a drug-induced perturbation.

Artificial neural networks (ANNs) are increasingly used to advance personalized medicine (*Alhusseini et al., 2020*; *Rogers, 2020*; *Sevakula et al., 2020*; *Jin et al., 2009*; *Trayanova et al., 2021*). Long-short-term-memory (LSTM)-based networks, which are capable of learning order dependence in sequence prediction problems (*Hochreiter and Schmidhuber, 1997*), have been widely used for cardiac monitoring purposes (*Guo et al., 2021*; *Shi et al., 2021*; *Picon et al., 2019*). They have been used to extract important biomarkers from raw ECG signals (*Ballinger, 2018*; *He et al., 2019*; *Hou et al., 2019*) and help clinicians to accurately detect common heart failure bio-markers in ECG screenings (*Ballinger, 2018*; *Warrick and Homsi, 2017*; *Oh et al., 2018*; *Chen et al., 2020*; *Wang and Zhou, 2019*; *Bian, 2019*). LSTM networks, which can catch existing temporal information in the electronic health records (EHRs), have been highlighted as the best predictive models using real-time data (*Maragatham and Devi, 2019*). LSTM-based classifiers have also empowered early arrhythmia detection by automatically classifying arrhythmias using ECG features (*Yildirim et al., 2019*; *Wang et al., 2019*; *Martis et al., 2013*; *Liu et al., 2019*; *Yildirim, 2018*). In addition, deep learning algorithms have been employed to predict drug-induced arrhythmogenicity associated with blockade of the delayed rectifier $K^+$ channel current ($I_{Kr}$) in the CMs encoded by human ether-à-go-go-related gene (hERG) (*Yang et al., 2020*) for sets of small molecules in drug discovery and screening process (*Yang et al., 2020*; *Cai et al., 2019*; *Zhang et al., 2019*; *Dickson et al., 2020*; *Ryu et al., 2020*; *Li et al., 2020*).

Here, we implemented a deep learning LSTM-based multitask network to classify iPSC-CM AP traces into drug-free and drugged categories and translate them into adult-CM AP waveforms. To collect robust realistic simulated data for training the multitask network, we paced simulated cardiac myocytes with the addition of a physiological noise current at matching cycle lengths for Kernik in silico iPSC-CMs (*Kernik et al., 2019*) and O'Hara–Rudy in silico human adult-CMs (*O'Hara et al., 2011*) to generate a population of drug-free simulated cardiac myocyte data. To ensure that our model could perform for both drug-free and drugged iPSC-CM and adult-CM APs simultaneously, we simulated drugged samples via both a simple drug-induced $I_{Kr}$ block model of hERG channel conduction, $G_{Kr}$, reduction by 1–50% and a complex Markov model of conformation-state dependent $I_{Kr}$ block in the presence of a clinical concentration, 2.72 ng/mL, of a potent hERG blocking drug dofetilide from our recent study (*Yang et al., 2020*). We evaluated the multitask network performance on a test dataset and showed excellent performance to translate and classify signals in the form of time-resolved AP traces. We performed ablation studies to reveal the most important iPSC-CM AP information for classifying AP traces into drug-free and drugged categories and network translation into adult-CM APs by removing iPSC-CM AP values during various time frames (feature ablation). We also explored the importance of individual LSTM network building blocks and how decoupling of the translation and classification tasks affected overall network performance. We then showed how the proposed multitask network can be applied even to scarce experimental data, which was also used to validate the model.

In this study, we show that developments in iPSC-CM experimental technology and cardiac electrophysiological modeling and simulation of iPSC-CMs can be leveraged for the application of ANNs as a universal approximator (*Goodfellow et al., 2016*) to find the most accurate mapping function that is capable of learning nonlinear relationships to predict disease phenotype and drug response in cardiac myocytes from immaturity to maturation.

## Results

In this study, we set out to build a multitask network that would perform two distinct tasks: the first task is to classify iPSC-CM APs into drug-free and drugged categories. The second goal is to translate iPSC-CM APs into corresponding adult-CM AP waveforms. To collect the data for training the multitask network, we simulated a population of 208 AP waveforms for both Kernik in silico human iPSC-CMs (*Kernik et al., 2019*; *Figure 1E*, blue) and O'Hara–Rudy in silico human adult-CMs (*O'Hara et al., 2011*; *Figure 1F*, blue). We ensured consistency across a population of simulated myocytes by applying physiological noise at matching the cycle lengths into the iPSC-CMs and adult-CMs. The cell variability in each population is intended to represent the individual variability that is observed in a drug-free human population (*Kernik et al., 2019*; *O'Hara et al., 2011*; *Tanskanen and Alvarez, 2007*). An average AP trace from the population is shown in *Figure 1A* for iPSC-CMs and *Figure 1B* for adult-CMs. In *Figure 1C, D*, the ionic currents underlying the in silico

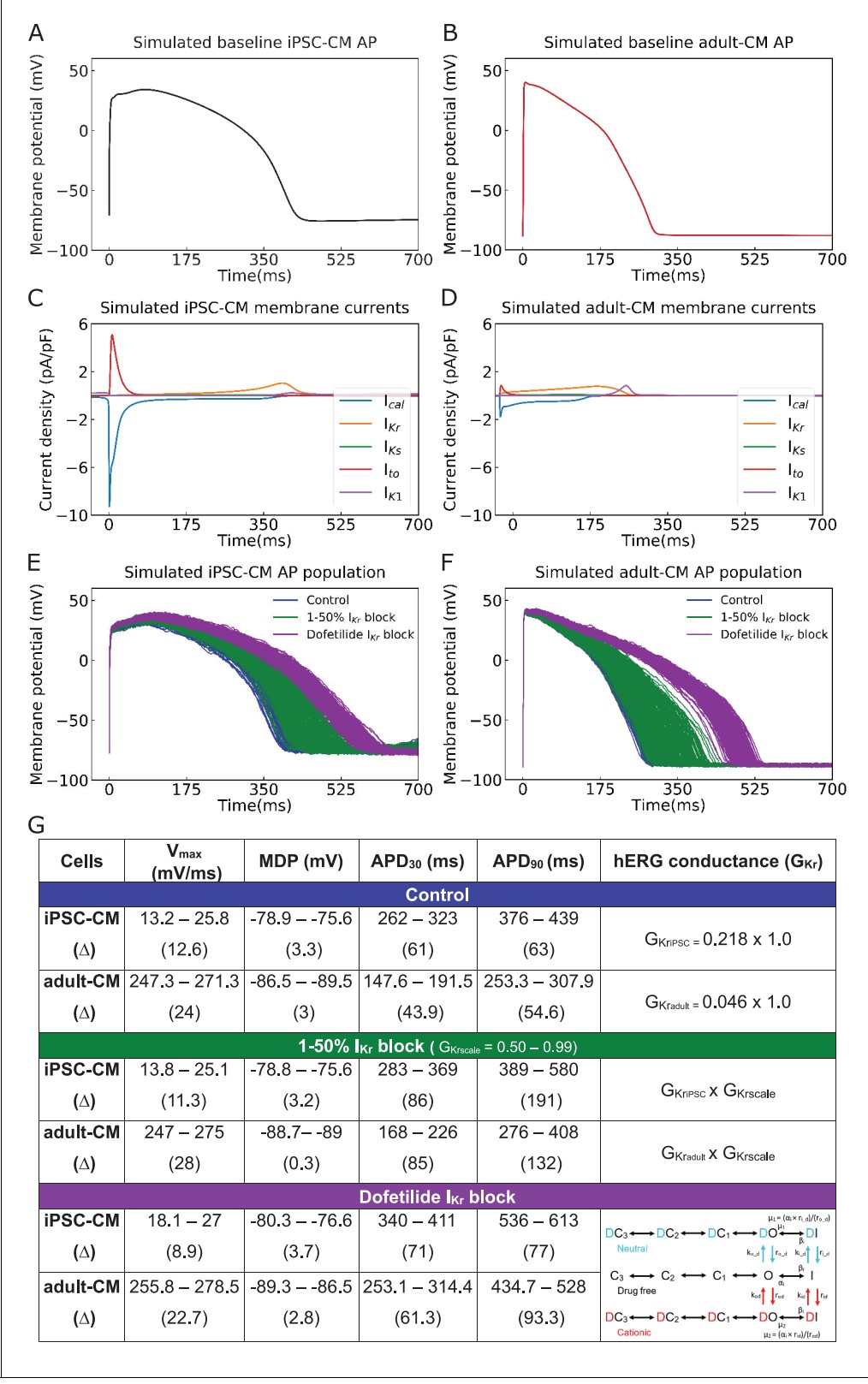

**Figure 1.** Cellular action potential (AP) and ionic currents for induced pluripotent stem cell-derived cardiomyocytes (iPSC-CMs) and adult cardiomyocytes (adult-CMs) (O'Hara–Rudy human ventricular APs). Comparison of cellular APs in the baseline model of (**A**) iPSC-CMs and (**B**) adult-CMs at a matched cycle length of 982 ms. (**C, D**) Simulated ionic current ($I_{CaL}$, $I_{Kr}$, $I_{Ks}$, $I_{to}$, $I_{K1}$) profiles during (**C**) iPSC-CM and (**D**) adult-CM APs. (**E**) APs of spontaneously beating iPSC-CM cells ($n = 208$) and (**F**) adult-CM APs at matched cycle lengths were simulated after incorporating physiological noise

*Figure 1 continued on next page*

Figure 1 continued

currents as drug-free (blue) and drugged $I_{Kr}$ modeled as simple $G_{Kr}$ reduction by 1–50% $I_{Kr}$ block (green) and a complex model of conformation-state dependent $I_{Kr}$ block in the presence of 2.72 ng/mL dofetilide (purple). (G) Comparison between iPSC-CM and adult-CM drug-free and drugged models with simple and complex $I_{Kr}$ block model schemes (as indicated in the right column), including upstroke velocity ($V_{max}$), maximum diastolic potential (MDP), and action potential duration (APD).

iPSC-CM APs and adult-CM APs show marked differences, one reason for the broadly expressed concerns about the applicability of utilizing immature iPSC-CM APs in the study of human disease and pharmacology. The substantial current differences illustrate the necessity of a generalized approach to perform translation from immature myocytes into mature myocytes. To ensure that our multitask network could perform over a range of conditions and model forms, we simulated drugged iPSC-CM and adult-CM APs via both a simple $I_{Kr}$ drug block model of $G_{Kr}$ reduction by 1–50% (250 samples in *Figure 1E, F*, green) and a complex model of conformation-state dependent $I_{Kr}$ block in the presence of 2.72 ng/mL dofetilide (300 samples in *Figure 1E, F*, purple). We combined the drug-free and drugged models with simple and complex $I_{Kr}$ block model schemes (758 samples) for training the multitask network. The differences in key parameters, upstroke velocity ($V_{max}$), maximum diastolic potential (MDP), and action potential durations (APD) across the three conditions are tabulated and shown in *Figure 1G*.

Next, we applied a digital forward and backward data filtering technique (*Gustafsson, 1996*) to the simulated iPSC-CM and adult-CM AP traces (*Figure 2*, left panels). Since we applied physiological noise to introduce a source of variability (as observed in human populations) in our model simulations, we assessed the possible phase distortion for AP waveforms following noise filtering. In *Figure 2* (right panels), the distribution of iPSC-CM and adult-CM AP duration at 90% repolarization ($APD_{90}$) values is shown. The near superimposition of the histogram distributions assures that noise filtering does not change the AP waveform morphology or time course and primarily removes existing vertical noises. *Figure 2A, B* shows simulated drug-free iPSC-CM and adult-CM APs and corresponding $APD_{90}$ distribution with physiological noise in blue and after applying the noise filtering technique in black for iPSC-CM APs and red for adult-CM APs. The same plots are illustrated for drugged AP traces with simple 1–50% $I_{Kr}$ block (*Figure 2C, D*) and with complex $I_{Kr}$ block model in the presence of 2.72 ng/mL dofetilide (*Figure 2E, F*). Next, we normalized drug-free and drugged noise-filtered iPSC-CM APs and adult-CM APs to use them as input and output, respectively, for training the multitask network.

The building blocks of the multitask network are illustrated in *Figure 3A*. The multitask network receives preprocessed simulation-generated iPSC-CM AP waveforms (noise-filtered and normalized) as input and scans whole AP time-series values through two stacked LSTM layers (*Figure 3A, D*). The LSTM layers remember the most important iPSC-CM AP values (features) they need to perform the translation and classification tasks and pass the information to two fully connected layers (*Figure 3A, E*), one for the translation task to predict the corresponding adult-CM AP waveform (*Figure 3B*) and one for the classification task to classify iPSC-CM APs into drug-free and drugged categories (*Figure 3C*).

The workflow for training and evaluating the multitask network is depicted in *Figure 4*. As described above, we generated simulated drug-free and drugged iPSC-CM and adult-CM APs and applied a noise filtering technique to the AP waveforms. The waveforms were then normalized in a data preprocessing step for more efficient training of the multitask network. We used preprocessed iPSC-CM APs as the network input and adult-CM APs along with corresponding drug-free and drugged labels as network outputs, respectively. Next, we randomly split input and output data in 70:10:20 ratio into three subcategories: training, validation, and test datasets. We used the training dataset for training the multitask network to simultaneously perform translation and classification. The mean squared error, $R^2$ score (*Devore, 2011*), and error in adult-CM $APD_{90}$ prediction were used as evaluation metrics for the translation task. For the classification task, area under the receiver operating characteristic (AUROC) curve (*Fawcett, 2006*), network prediction accuracy, precision, and recall (*Powers, 2011*) were used to evaluate the network performance. To prevent overfitting, we calculated the evaluation metrics for both tasks using validation data during each iteration of training and compared those with values from the training dataset. When the model performance on the training dataset exhibited degradation relative to the validation dataset, we ceased training and

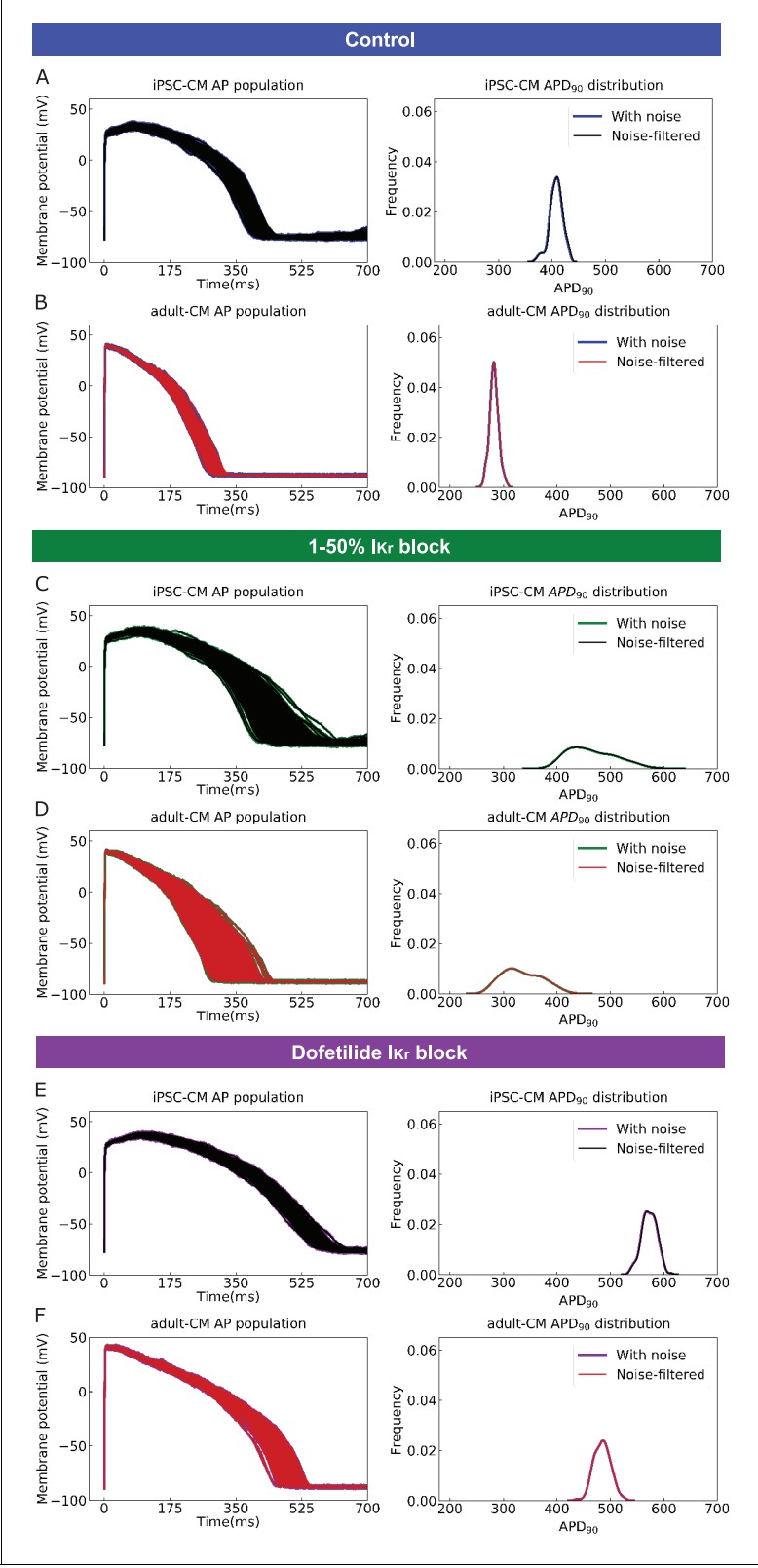

**Figure 2.** Application of a digital forward and backward data filtering technique to simulated induced pluripotent stem cell-derived cardiomyocyte (iPSC-CM) and adult cardiomyocyte (adult-CM) action potentials (APs) population (left panels) indicates zero phase distortion for $APD_{90}$ value distributions (right panels). (**A**) drug-free iPSC-CM APs with physiological noise in blue and after applying the noise filtering technique in black; (**B**) drug-free adult-CM APs – blue and red traces; (**C**) drugged iPSC-CM APs with 1–50% $I_{Kr}$ block – green and black traces; (**D**) drugged adult-

*Figure 2 continued on next page*

*Figure 2 continued*

CM APs with 1–50% $I_{Kr}$ block – green and red traces; (**E**) drugged iPSC-CM APs with 2.72 ng/mL dofetilide – purple and black traces; and (**F**) drugged adult-CM APs with 2.72 ng/mL dofetilide –purple and red traces.

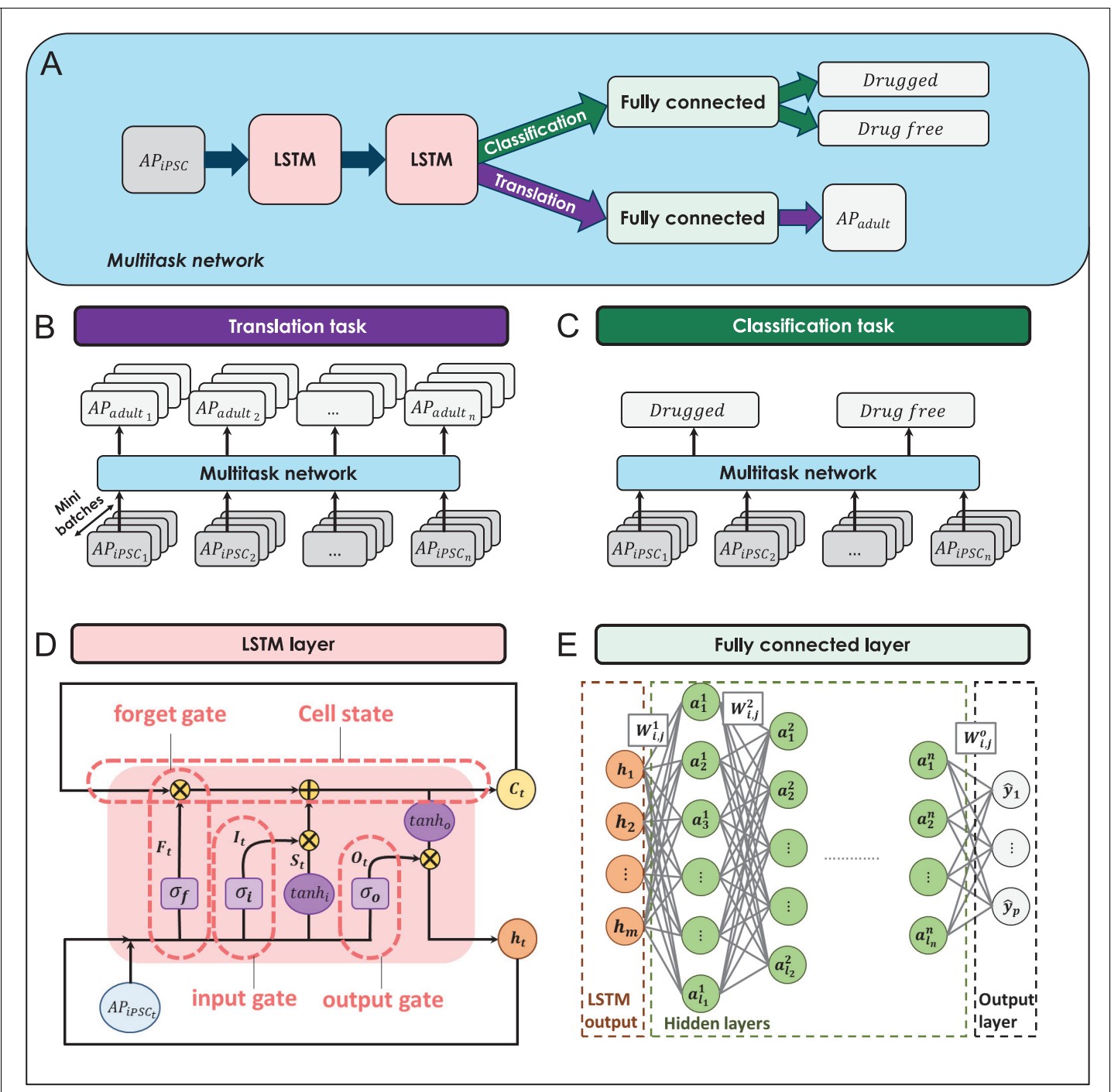

**Figure 3.** The building blocks of the multitask network. (**A**) The general overview of the multitask network presented in this study. (**B**) The translation task to reconstruct adult cardiomyocyte (adult-CM) action potentials (APs) from the corresponding induced pluripotent stem cell-derived cardiomyocyte (iPSC-CM) APs. (**C**) The classification task to classify iPSC-CM APs into drug-free and drugged categories. (**D**) The logic flow process in the long-short-term-memory (LSTM) layers. (**E**) The architecture of the implemented fully connected layers in the multitask network.

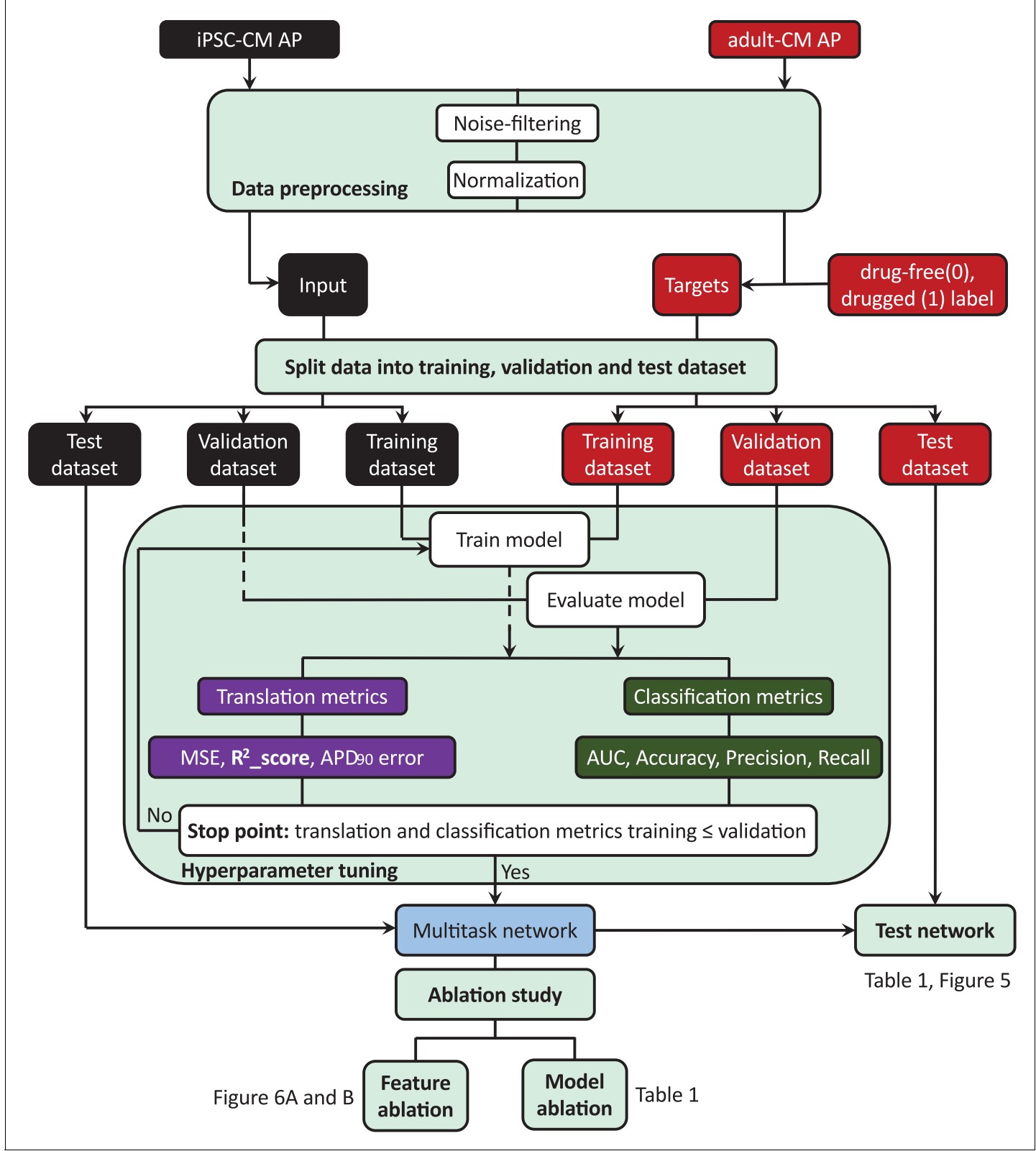

**Figure 4.** Machine learning workflow in this study. (1) Data preprocessing includes noise filtering and normalization of the drug-free and drugged induced pluripotent stem cell-derived cardiomyocyte (iPSC-CM) and adult cardiomyocyte (adult-CM) action potentials (APs). (2) Incorporating the preprocessed iPSC-CM APs as input and adult-CM APs and corresponding labels (drug-free [0] and drugged [1]) of iPSC-CM APs as targets into the multitask network. (3) Splitting the input and target data into training, validation and test set, and using training and validation set for training and

*Figure 4 continued on next page*

*Figure 4 continued*

tuning the network hyperparameters. (4) Comparing the network performance for training set and validation set to decide when to stop training and tuning the network hyperparameters. (5) Testing the overall multitask network performance using holdout test dataset and removing the long-short-term-memory (LSTM) layers, classification task (model ablation), and iPSC-CM AP values at different time frames (feature ablation) to study the performance of the network in the absence of its building blocks.

tuning of the network hyperparameters. We evaluated the underlying mechanisms that inform the network performance by using a holdout test dataset to perform an ablation study. The ablation study allowed us to identify the most important information for network performance and is an indicator of the data that the network deems most important to remember to classify AP traces into drug-free and drugged categories and allow accurate translation into adult-CM APs (feature ablation). Finally, we performed a type of network component dissection by sequentially eliminating individual LSTM layers or the classification task to determine if all elements of the network are important to the overall performance.

*Figure 5* and *Table 1* illustrate the overall multitask network performance for translation and classification tasks for the training and test datasets. *Figure 5A, B* represents iPSC-CM APs (black), which were used for training and testing the multitask network, respectively. *Figure 5B, E* depicts the comparison between simulated (red) and translated (cyan) adult-CM APs used for the training and testing the network. The comparison between histogram distribution of $APD_{90}$ values for simulated and translated adult-CM APs in *Figure 5C, F* shows good agreement in terms of the frequency of virtual cells with similar APD.

The performance evaluation metrics for both the translation and classification tasks are listed in *Table 1*. The multitask network exhibits high accuracy in performing translation, despite large variability in APDs and regardless of the underlying model form. The network is able to translate iPSC-CM APs into adult-CM APs with less than 0.003 mean-squared error (MSE), 0.99 $R^2$ score, and <4% error in $APD_{90}$ prediction for both training and test datasets. To evaluate the network performance

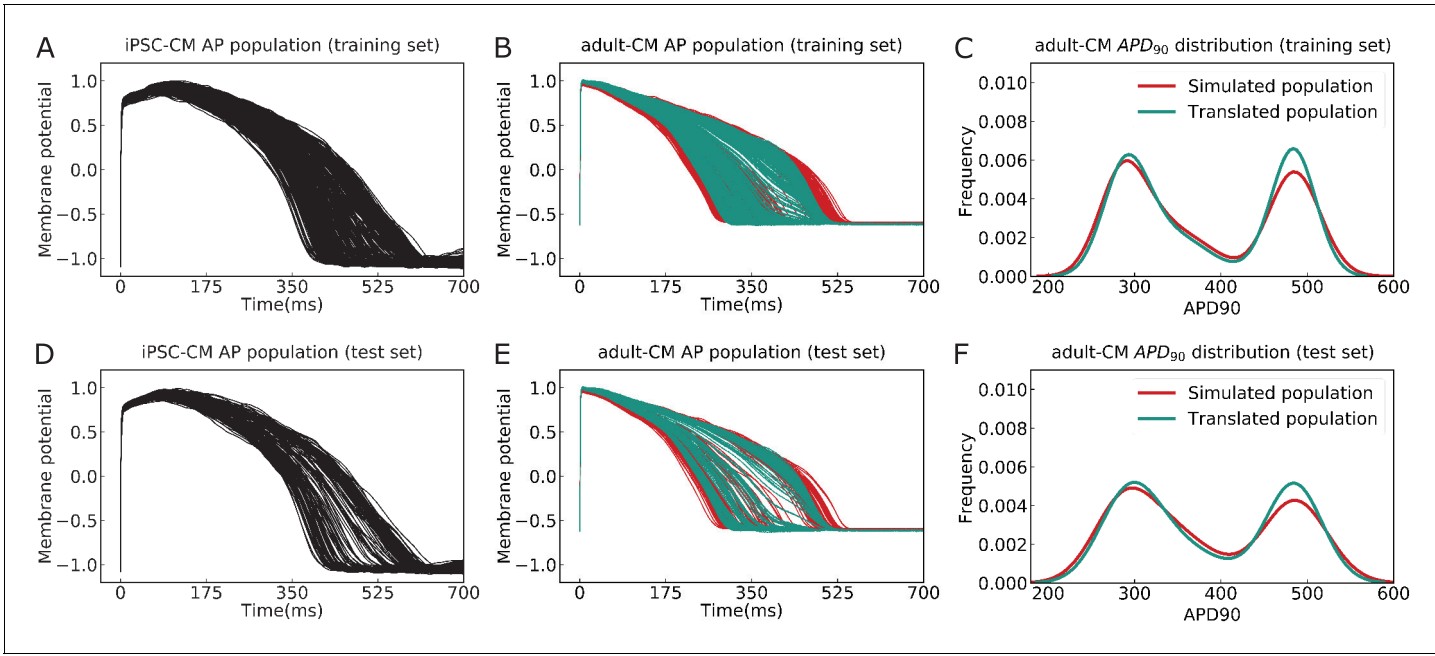

**Figure 5.** The performance of the multitask network for translating induced pluripotent stem cell-derived cardiomyocyte (iPSC-CM) action potentials (APs) into adult cardiomyocyte (adult-CM) APs. (**A**) The iPSC-CM APs used for training the multitask network contained a variety of drug-free and drugged action potential morphologies (training set). (**B**) Comparison between simulated (red) and translated adult-CM APs (cyan) in the training set. (**C**) Comparison between the histogram distribution of $APD_{90}$ values for simulated and translated adult-CM APs in the training set. (**D**) Dedicated iPSC-CM APs for testing the performance of the multitask network (test set). (**E**) Comparison between simulated (red) and translated adult-CM APs (cyan) in the test set. (**F**) Comparison between histogram distribution of $APD_{90}$ values for simulated and translated adult-CM APs in the test set.

**Table 1.** Statistical measures for evaluating the performance of the multitask network for both iPSC-CM AP trace classification into drug-free and drugged categories and their translation into adult-CM APs for training and test datasets as well as the effect of removing LSTM layers and classification task on the network performance.

**Translation**

| Performance metrics | MSE | $R^2$ score | Error in $APD_{90}$ prediction |
|---|---|---|---|
| Training dataset | 0.0027 | 0.992 | 3.41% |
| Test dataset | 0.0029 | 0.991 | 3.60% |
| Remove LSTM layers test dataset | 0.0031 | 0.991 | 3.78% |
| Remove classification task test dataset | 0.0034 | 0.990 | 4.33% |

**Classification**

| Performance metrics | AUROC | Accuracy | Recall | Precision |
|---|---|---|---|---|
| Training dataset | 0.93 | 92% | 0.92 | 0.93 |
| Test dataset | 0.91 | 92% | 0.92 | 0.92 |
| Remove LSTM layers test dataset | 0.90 | 92% | 0.90 | 0.91 |

iPSC-CM: induced pluripotent stem cell-derived cardiomyocyte; AP: action potential; adult-CM: adult cardiomyocyte; AP: action potential; AUROC: area under the receiver operating characteristic curve; LSTM: long-short-term-memory.

for the classification task, we compared the AUROC, prediction accuracy, recall, and precision for both training and test datasets. The multitask network proved to perform well in categorizing iPSC-CM APs into drug-free and drugged waveforms with approximately 90% accuracy (*Table 1*). Finally, we performed a type of network component dissection by sequentially eliminating individual LSTM layers or the classification task to determine if all elements of the network are important to the overall performance. The impact of removing these elements of the network on the network performance is shown in *Table 1*.

Next, we performed a 'computational' ablation study as a correlate to the types of physiological ablations that are used to examine the roles and functions of a physiological system (*LeCun et al., 1989*; *Reale et al., 1987*). We tested how the performance of the multitask network would change by removing various information contained within specified time frames as shown in *Figure 6A, B*. To reveal the most important iPSC-CM AP information for classifying iPSC-CM APs into drug-free and drugged traces and translation into adult-CM APs, we did not allow the network to process data from within designated time frames from the iPSC-CM APs (feature ablation). We then retrained the multitask network by setting the missing information equal to zero and compared the calculated AUROC for classification task and MSE in adult-CM APs translation (red bars) with the recorded corresponding values for multitask network (green line) when it was provided full access to the complete iPSC-CM AP data. We observed that the network is extremely sensitive to information contained within the 400–500 ms time frame (blacked dashed bar in *Figure 6A, B*).

This result suggests that the most important information needed to classify iPSC-CM APs into drug-free and drugged traces and distinguish adult-CM AP signals from iPSC-CM AP signals is contained in a particular region of the AP plateau. The time frame of the AP between 400 and 500 ms (*Figure 6A, B*) corresponds to a phase of exquisite sensitivity to perturbation. We have identified this particular AP range in an earlier study as the phase when the membrane resistance of the myocyte increases markedly (*Figure 6C*; *Yang et al., 2015*). This occurs as the inward and outward currents balance each other, leading to a net whole cell current that is nearly constant so that $dI \rightarrow 0$, $dV/dI \rightarrow \infty$ (*Figure 6D*), followed by a rapid reduction in outward current. *Figure 6E* demonstrates that individual current densities have a period of inward and outward current balance followed by rapid changes in $I_{Kr}$ and other repolarizing currents at 400–500 ms time interval.

We next set out to demonstrate the real-world utility of the multitask classification and translation network by applying the network to experimental data. We used experimental iPSC-CM APs from the Kurokawa lab (*Figure 7A*) as the input data into the multitask network and translated to predicted adult-CM APs as shown in *Figure 7B*. The translation notably resulted in a reduction in variability in APD in the adult translated cells, consistent with our simulated results and with previous experimental observations (*Blinova et al., 2018*; *Fabbri et al., 2019*). In an additional validation of

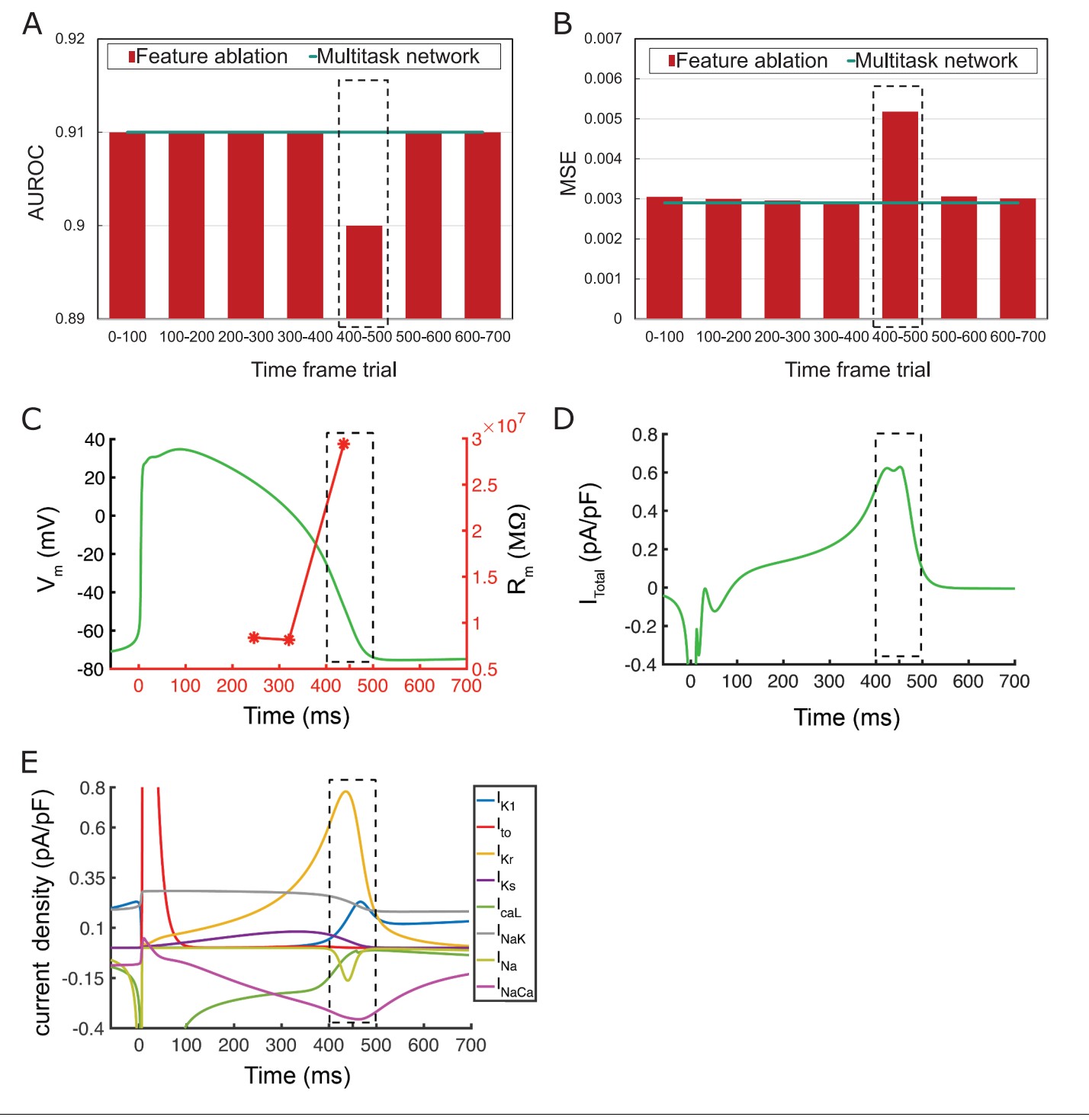

**Figure 6.** The feature ablation study on the proposed multitask network is performed by removing induced pluripotent stem cell-derived cardiomyocyte (iPSC-CM) action potential (AP) values during different time frames and evaluating their importance on drug-free and drugged iPSC-CM AP classification and adult cardiomyocyte (adult-CM) AP translation. The largest effect (most important information) is observed at 400–500 ms interval (dotted black line). (**A**) Comparison between intact multitask network area under the receiver operating characteristic curve (AUROC) and obtained AUROC values for drug-free and drugged iPSC-CM AP classification during removal of indicated time frames within iPSC-CM APs. (**B**) Comparison between intact multitask network mean-squared error (MSE) (cyan line) and obtained MSE values for adult-CM AP translation during removal of indicated time frames within iPSC-CM APs (red bars). (**C**) AP trace (green) and membrane resistance (red) as a function of simulation time indicating very high values (as dI → 0, dV/dI → ∞) for the latter at 400–500 ms. (**D**) Total current density, I$_{total}$, demonstrates a plateau followed by a rapid decline

*Figure 6 continued on next page*

*Figure 6 continued*

at 400–500 ms. (E) Individual current densities indicate a period of inward and outward current balance followed by rapid changes in $I_{Kr}$ and other repolarizing components at 400–500 ms time interval.

the multitask network, we undertook a test of the network to accurately translate drug block in iPSC-CMs to adult AP effects and then compared the predicted results with measured experimental data (*O'Hara et al., 2011*). We first simulated iPSC-CM APs with 50% block of $I_{Kr}$. We then used these simulated APs as an input for the multitask network and used the output from the translation task to predict 50% block on adult-CMs. In *Figure 7C*, the translated drugged APD$_{90}$ values are shown as turquoise asterisks from spontaneously beating (~1000 ms cycle length) simulated iPSC-CMs plotted against simulations from O'Hara–Rudy adult-CM APs with 50% $I_{Kr}$ block (red curve) and experimental 50% block of $I_{Kr}$ by *1* μM E-4031 (blue squares) (*O'Hara et al., 2011*). These data validate that the effects of drug block in iPSC-CMs can be successfully translated to predict drug effect on adult human cardiomyocyte APs.

## Discussion

In this study, we developed a data-driven deep learning approach to address the well-known shortcomings in the iPSC-CM platform. A concern with iPSC-CM is that the data collection results in measurements from immature APs, and it is unclear if these data reliably indicate impact in the adult cardiac environment (*Navarrete et al., 2013*; *Casini et al., 2017*; *Goversen et al., 2018*; *Knollmann, 2013*; *Sinnecker et al., 2013*; *Blinova et al., 2017*). Here, we set out to demonstrate a new way to allow translation of results from the iPSC-CM to a mature adult cardiac response. The deep learning network also revealed new mechanisms that are critical to convert iPSC-CM APs to mature adult cardiac APs.

Application of a deep learning ANN to simultaneously translate and classify signals from simulated iPSC-CMs for both drug-free and drugged conditions has several key advantages. Because there is no need for the multitask network user to a priori define the important system parameters, the approach is by definition an unbiased model. A key part of the 'artificial intelligence' is learning from the data to make decisions about which elements of the data are the most important. Another benefit is the model-agnostic approach in that the learning network is indifferent to the underlying form of the models and can readily translate time-series data from any source. The nonlinearity of the system is accepted by the deep learning approach, and there are no assumptions made about cardiac protein expression levels and changes in their function during cardiomyocyte maturation. The deep learning ANN can successfully translate simple pore block and complex conformation state-dependent channel–drug interaction models. The network can learn from multiple sources of data even when they are generated from different models and learns from all the data sources concurrently for robust and successful translation. All of these aspects of the technology presented here suggest broad applicability for use across ages, species, and conditions, and we demonstrate its utility for translating and predicting experimental data.

The multitask network presented here performed well in the setting of the noted variability in measurements from iPSC-CM APs. As described in *Figure 1*, we utilized a modeling and simulation approach from our recent studies (*Kernik et al., 2019*; *Kernik et al., 2020*) to generate a population of iPSC-CM APs that incorporate variability comparable to that in experimental measurements. Utilizing simulated data presented a unique opportunity: we were able to generate large amounts of data that were used both to train and optimize the network and then to test the network with specifically designated distinct simulated datasets. Utilizing simulated data to train a deep learning network may constitute a widely applicable approach that could be used to train a variety of networks to perform multiple functions where access to comparable experimental data is not feasible.

The multitask network exhibits high accuracy in performing translation, despite large variability in APDs and regardless of the underlying model form (*Figure 5* and *Table 1*). The network was able to translate iPSC-CM APs into adult-CM APs with less than 0.003 MSE, *0.99* R$^2$ score, and less than 4% error in APD$_{90}$ prediction for both the training and test datasets. To evaluate the network performance for the classification task, we compared the AUROC, prediction accuracy, recall, and precision for both training and test datasets. The multitask network proved to perform well in

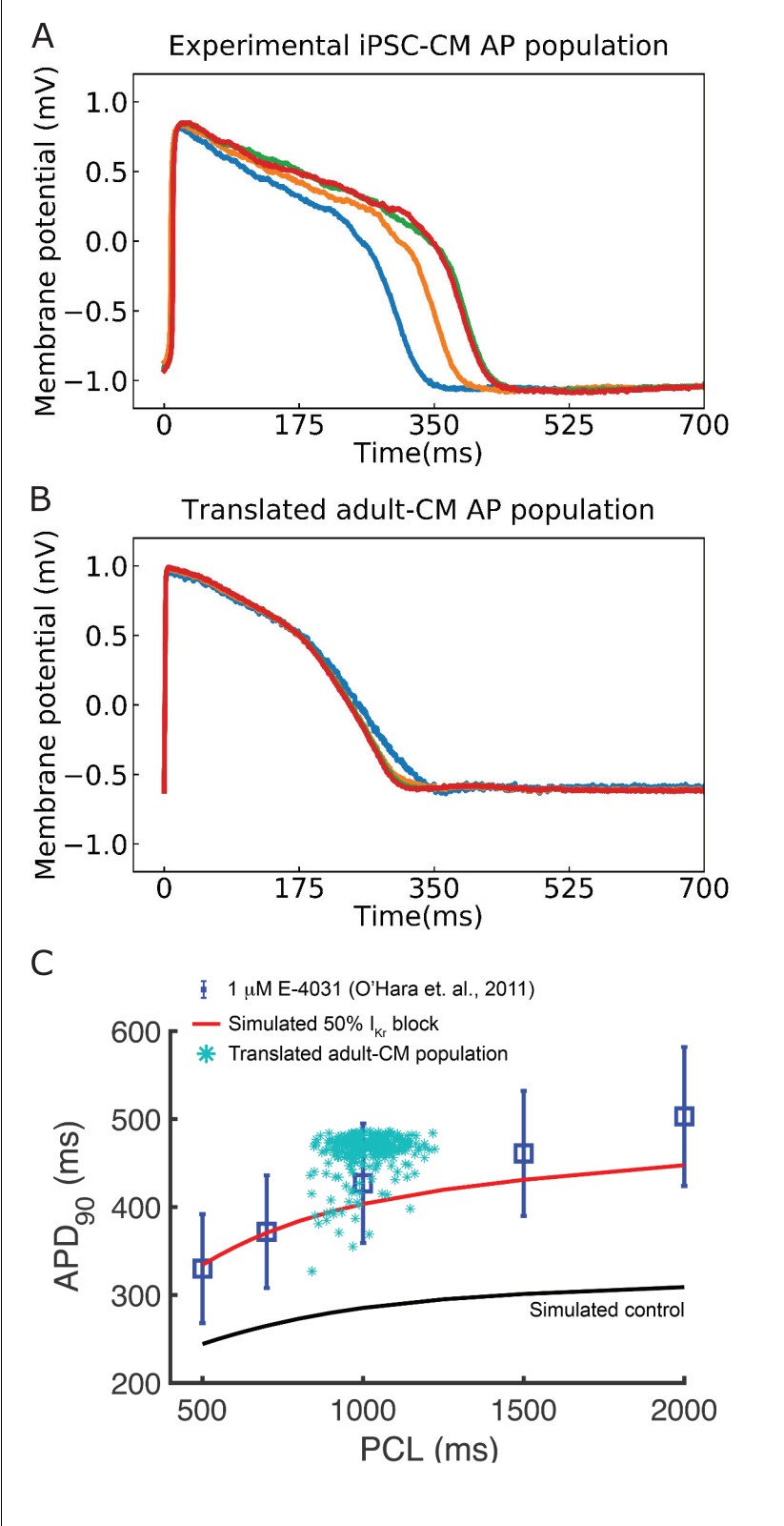

**Figure 7.** Translation of experimentally recorded induced pluripotent stem cell-derived cardiomyocyte (iPSC-CM) action potentials (APs) into adult cardiomyocyte (adult-CM) APs to demonstrate the deep learning network applied to experimental data and validate the multitask network performance. (**A**) Experimentally recorded iPSC-CM APs from the Kurokawa lab. (**B**) Translated adult-CM APs from experimentally recorded iPSC-CM APs via the multitask network. (**C**) Comparing translated adult-CM $APD_{90}$ values with 50% $I_{Kr}$ block from spontaneously beating simulated iPSC-CMs

*Figure 7 continued on next page*

*Figure 7 continued*

around 1000 ms beating frequency (turquoise asterisks) with previously published simulated (red curve for drugged and black for drug-free control) and experimental (blue squares) values from the O'Hara–Rudy study (*Shaheen et al., 2018*) indicates that predictions fall within the range of experimentally reported values at 1 Hz.

categorizing iPSC-CM APs into drug-free and drugged waveforms with approximately *90%* accuracy (*Table 1*). Finally, we performed a type of network component dissection by sequentially eliminated individual LSTM layers or the classification task to determine if all elements of the network are important to the overall performance. The impact of removing these elements of the network on its performance is shown in *Table 1*. The studies show that the multitask network conferred additional benefit over considering the translation task alone. For example, we noted that adding the classification task to distinguish drug-free and drugged APs could improve the performance of the translation task (*Table 1*).

When we performed an ablation study to prevent the deep learning network from using information within prespecified time windows, the results revealed that the most important information needed to classify iPSC-CM APs into drug-free and drugged traces and predict adult-CM APs from iPSC-CM AP signals is contained in the phase of the AP between 400 and 500 ms (*Figure 6*). This result suggests that the most important information needed to classify and distinguish iPSC-CM AP signals from adult-CM AP signals is contained in the range of the AP that corresponds to a phase of exquisite sensitivity to perturbation. We have identified this particular AP range in an earlier study as the phase when the membrane resistance of the myocyte increases markedly (*Figure 6C*; *Yang et al., 2015*). This occurs as the inward and outward currents balance each other, leading to a net whole cell current that is unchanging (dI $\rightarrow$ 0, dV/dI $\rightarrow$ $\infty$), followed by a rapid reduction in the outward current (*Figure 6D, E*). It is not surprising that this time frame is shown to contain the most important information to perform the classification task as the effect of $I_{Kr}$ block is critical during the high resistance phase of the membrane potential. It is possible that other types of perturbations (e. g., Na channel blocker, ischemia) may lead to a different outcome, and we will pursue those questions in future studies.

Following the optimization and demonstration of the network as an accurate tool for both translating and classifying data, we then used the same network to translate experimentally obtained data. We showed that the proposed network can effectively take experimental data as an input from immature iPSC-CM APs and translate those data to produce adult AP waveforms. It is notable that the variation observed in the adult-CM AP duration is smaller compared to iPSC-CM APDs (*Figure 7A, B*). This has been observed both experimentally (*Blinova et al., 2018*; *Fabbri et al., 2019*) and in our simulated cell environment (*Kernik et al., 2019*; *Kernik et al., 2020*). Although the simulated iPSC-CM has a large initial calcium current (*Figure 1C*) compared to the simulated adult-CM (*Figure 1D*), the amplitude of currents flowing through adult-CM AP plateau is notably larger. The immature iPSC-CM cells have low conductance during the AP plateau, rendering it comparably higher resistance. For this reason, small perturbations to the iPSC-CM APs have a larger impact on the resulting AP duration than observed in adult cells (*Yang et al., 2015*). We also used simulated iPSC-CM APs subject to 50% block of $I_{Kr}$. We translated those data to adult-CM APs and then compared with the previously reported impact of 50% $I_{Kr}$ block on adult human cell APs from experiments (*O'Hara et al., 2011*) and noted excellent agreement, thereby providing validation of our network.

The deep learning algorithm presented here has the benefit of automating feature extraction without any predetermination of the feature. It also allows for the translation of time-course data from simulated or experimental datasets. However, there are some limitations to the approach. One limitation is the requirement for multiple datasets that are of sufficient quality for training – the more robust the training set, the higher the accuracy of the task. It is possible that this limitation can be addressed in future studies by developing new methods for data extraction and data interpolation in sparse datasets. We addressed this limitation by utilizing simulated data to train the network, and this approach might be applicable for a variety of physiological problems. Simulated data can be generated to constitute a robust dataset that can be used to train the multitask model and allow extraction of the relevant features from any time-course dataset.

In this study, we show that a deep learning network can be applied to classify cells into the drug-free and drugged categories and can be used to predict the impact of electrophysiological perturbation across the continuum of aging from the immature iPSC-CM AP to the adult ventricular myocyte AP. By extension, the classification task might even be applied to distinguish cellular-level signals derived from cells cultured using different protocols. We translated experimental immature APs into mature APs using the proposed network and validated the output of some key model simulations with experimental data. The multitask network in this study was used for translation of iPSC-CMs to adult APs but could be readily extended and applied to translate data across species and classify data from a variety of systems. Also, another extension of the technology presented here is to predict the impact of naturally occurring mutations and other genetic variations (*Yoshinaga et al., 2019*).

## Materials and methods

### Simulated data for training and testing the multitask network
#### The drug-free iPSC-CM and adult-CM APs
The Kernik in silico iPSC-CM baseline cells were paced from resting steady state. The O'Hara–Rudy in silico endocardial cell model was used for the baseline adult-CMs (*O'Hara et al., 2011*). The control adult-CMs were paced at the cycle length of 982 ms to match the cycle length of the last beat of the spontaneously depolarizing iPSC-CM AP. The iPSC-CM AP populations ($n$ = 208) were generated by incorporating physiological noise (see Simulated physiological noise currents section below). The adult-CMs were paced with noise for 100 beats after reaching steady state at the matching cycle length of the last beat of iPSC-CM AP populations. The numerical method used for updating the voltage was Forward Euler method (*Atkinson, 2008*).

### A simple drug-induced 1–50% $I_{Kr}$ block model through $G_{Kr}$ reduction
The iPSC-CMs and the adult-CMs populations were paced with 1–50% $I_{Kr}$ block with 1% increments. This was accomplished by scaling down hERG channel ($I_{Kr}$) conduction, $G_{Kr}$, by the fraction of the block, $G_{Krscale}$, in the 0.50–0.99 range with 0.01 decrements (see central rows in *Figure 1G*). The adult-CM model was simulated at five varying beating rates for each percentage of block that matches to the last beat of iPSC-CMs with 1–50% $I_{Kr}$ block ($n$ = 250). For example, one drugged adult-CM (50% $I_{Kr}$ inhibition) was paced at cycle length of 1047 ms to match the cycle length of the last beat of iPSC-CMs AP with 50% $I_{Kr}$ block.

### Complex model of conformation-state dependent $I_{Kr}$ block in the presence of 2.72 ng/mL dofetilide
The $I_{Kr}$ channel Hodgkin–Huxley model in both iPSC-CM and adult-CM AP models was replaced with a drug–hERG channel interaction Markov model (see bottom rows in *Figure 1G*) that we have previously published (*Yang et al., 2020*). iPSC-CM ($n$ = 300) and adult-CM AP populations ($n$ = 300) were generated with physiological noise in the presence of 2.72 ng/mL dofetilide, a potent hERG channel blocker. The adult-CM populations were paced with dofetilide for 100 beats after reaching steady state at the matching cycle length of the last beat of iPSC-CM AP populations with dofetilide as described above. The simulated drugged and drug-free iPSC-CM and adult-CM AP data used for training and testing the multitask network have been made publicly available at Clancy lab GitHub. (https://github.com/ClancyLabUCD/Multitask_network/tree/master/data, copy archived at swh:1:rev: 7f2b653a91f552d66ae2d9b70b720f8706b36da3, *Aghasafari, 2021*).

### Simulated physiological noise currents
Simulated noise current was added to the last 100 paced beats in the simulated AP models, and simulated APs were recorded at the 2000th paced beat in single cells. This noise current was modeled using the equation from *Tanskanen and Alvarez, 2007*

$$V_{t+t} = V_t - \frac{I(V_t)t}{C_m} + \xi n \sqrt{t} \qquad (1)$$

where $n \in N(0,1)$ is a random number from a Gaussian distribution, and $\Delta t$ is the time step. $\xi = 0.3$ is the diffusion coefficient, which is the amplitude of noise. The noise current was generated and applied to membrane potential, $V_t$, throughout the last 100 beats of simulated time course.

## Experimental iPSC-CMs

Human iPSC-CMs (201B7, RIKEN BRC, Tsukuba, Japan) were cultured and subcultured on *SNL76/7* feeder cells as described in detail previously (*Li et al., 2017*). Cardiomyocyte differentiation was performed as described (*Li et al., 2017*). Commercially available iCell-cardiomyocytes (FUJIFILM Cellular Dynamics, Inc, Tokyo, Japan) were cultured according to the manual provided from the company. APs were recorded with the perforated configuration of the patch-clamp technique as described in detail previously (*Li et al., 2017*). Measurements were performed at $36 \pm 1°C$ with the external solution composed of (in mM) NaCl (135), $NaH_2PO_4$ (0.33), KCl (5.4), $CaCl_2$ (1.8), $MgCl_2$ (0.53), glucose (5.5), and HEPES, pH 7.4. To achieve patch perforation (10–20 M$\Omega$; series resistances), amphotericin B (0.3–0.6 µg/mL) was added to the internal solution composed of (in mM) aspartic acid (110), KCl (30), $CaCl_2$ (1), adenosine-5′-triphosphate magnesium salt (5), creatine phosphate disodium salt (5), HEPES (5), and EGTA (11), pH 7.25. In quiescent cardiomyocytes, APs were elicited by passing depolarizing current pulses (2 ms in duration) of suprathreshold intensity (120% of the minimum input to elicit APs) with a frequency at 1 Hz unless noted otherwise. The experimental data used for the model validation have been made publicly available at Clancy lab GitHub. (https://github.com/ClancyLabUCD/Multitask_network/blob/master/data/clean_data/experiments.csv).

## The multitask network architecture

The multitask network comprised two stacked LSTM layers followed by independent fully connected layers (*Figure 3A*) for the classification and translation tasks. The LSTM layers memorized the important information the network needed to perform two discussed tasks and then transferred the extracted information (features) into the subsequent fully connected layers to translate iPSC-CM APs into adult-CM AP waveforms (*Figure 3B*) and classify iPSC-CM APs into drug-free and drugged categories (*Figure 3C*).

## LSTM layers (*Figure 3D*)

We used LSTM layers as the first two layers of the multitask network to promote network temporal information learning which data in a sequence was important to keep or to throw away. At each time step, the LSTM cell took in three different pieces of information, the current input data $(AP_{iPSC_t})$, incoming short-term memory (hidden state) $(h_{t-1})$ and incoming long-term memory (cell state) $(C_{t-1})$. The LSTM layers were responsible for extracting the most important information while scanning the AP traces using the short- and long-term memory components. The short-term memory weighted the importance of AP values at subsequent time steps and long-term memory has been using the short-term memory to decide the overall importance of all AP values from the beginning ($t = 0$ ms) to the end ($t = 701$ ms) for performing classification and translation tasks. The LSTM cells contained internal mechanisms called gates. The gates were neural network with weights ($w$) and bias terms ($b$) that regulated the flow of information at each time step before passing on the long-term and short-term information to the next cell (*Cheng et al., 2016*). These gates are called input gate, forget gate, and output gate (*Figure 3D*).

The forget gate, as the name implies, determined which information from the long-term memory should be kept or discarded. This was done by multiplying the incoming long-term memory by a forget vector generated by the current input $(AP_{iPSC_t})$ and incoming short-term memory $(h_{t-1})$. To obtain the forget vector, the incoming short-term memory and current input were passed through a sigmoid function $(\sigma_f)$ (*Olah, 2017*). The output vector of sigmoid function, $F_t$, (*Equation 2*) was a binary comprising 0s and 1s and was then multiplied by the incoming long-term memory $(C_{t-1})$ to choose which parts of the long-term memory were retained.

$$F_t = \sigma_f \left( w_f AP_{iPSC_t} + w_f h_{t-1} + b_f \right) t \in \{0, 1, \ldots, 701\} \tag{2}$$

The input gate decided what new information is being stored in current long-term memory $(C_t)$. It considered the current input $(AP_{iPSC_t})$ and the incoming short-term memory $(h_{t-1})$ and transformed the values to be between 0 (unimportant) and 1 (important) using a sigmoid activation function $(\sigma_i)$

(*Equation 3*). The second layer in input gate took the incoming short-term memory ($h_{t-1}$) and current input ($AP_{iPSC_t}$) and passed them through a hyperbolic tangent activation function ($tanh_i$) to regulate the network computation (*Equation 4*).

$$I_t = \sigma_i(W_i AP_{iPSC}t + W_i h_{t-1} + b_i) t \in \{0, 1, \ldots, 701\} \tag{3}$$

$$S_t = tanh_i(w_s AP_{iPSC}t + w_s h_{t-1} + b_s) \tag{4}$$

The outputs from the forget and input gates then underwent a pointwise addition to find the current long-term memory ($C_t$) (*Equation 5*), which was then passed on to the next cell.

$$C_t = F_t * C_{t-1} + I_t * S_t \tag{5}$$

Finally, the output gate utilized current input ($AP_{iPSC_t}$) and the incoming short-term memory ($h_{t-1}$) and passed them into a sigmoid function ($\sigma_o$) (*Equation 6*). Then the current long-term memory ($C_t$) passed through a tanh activation function ($tanh_o$) and the outputs from these two processes were multiplied to produce the current short-term memory $h_t$ (*Equation 7*).

$$O_t = \sigma_o(w_o AP_{iPSC}t + w_o h_{t-1} + b_o) \tag{6}$$

$$h_t = O_t * tanh_o(C_t) \tag{7}$$

The short-term and long-term memory produced by these gates were carried over to the next cell for the process to be repeated. The output of LSTM layers for each time step ($h_t$) was obtained from the short-term memory, also known as the hidden state, and was subsequently passed into fully connected layers to perform the translation and classification tasks as described below.

## Fully connected layers (*Figure 3E*)

The fully connected neural network layers contained input, hidden, and output layers (*Figure 2E*) with various numbers of neurons ($l_r$). Every neuron in a layer was connected to neurons in the next layer (*Krogh, 2008*). Fully connected layers received the output of LSTM layers as input. The fully connected layers calculated a weighted sum of LSTM outputs and added a bias term to the outputs. These data were then passed to an activation function (*f*) to define the output for each neuron (*Equations 8 and 9*; *Carugo and Eisenhaber, 2010*).

$$a_j^k = f\left(Z_j^k\right) \tag{8}$$

$$Z_j^k = W_{i,j}^k * a_j^{k-1} + b^k \tag{9}$$

where $k \in \{1, \ldots, n\}$ and $(i, j)$ represent the number of hidden layers and neurons in each pair of subsequent hidden layers ($l_r, l_{r+1}$). The optimized values for these parameters were found via hyperparameter tuning where $a^k$ is each neuron output. $a^0 \in \{h_1, \ldots, h_m\}$ is the LSTM layer output and the input to the fully connected layers, and $a^{n+1}$ is the network output: $y \epsilon \{y_{t_i}, y_{c_i}\}$, where $y_{t_i}$ and $y_{c_i}$ are the outputs for translation and classification tasks, respectively. We first assigned random values to all network parameters $\theta_t$; each neuron weight ($W_{i,j}$) (*Figure 3E*), bias term ($b$), which is a constant added to calculate the neurons output and other network hyperparameters (the number of hidden layers, the number of neurons for each hidden layer and activation functions for each hidden layer) to start the optimization process for finding the best network infrastructure. Next, we estimated the network errors using MSE (*Equation 10*) and cross-entropy loss functions (*Equation 11*) to map the translation and classification tasks (*Goodfellow et al., 2016*; *Murphy, 2012*), respectively.

$$MSE = \frac{1}{m} \sum_{i=1}^{n} \left\| y_{t_i} - y_{t_i} \right\|^2 \tag{10}$$

$$CrossEntropy = -\left(y_{c_i} \log(y_{c_i}) + (1 - y_{c_i}) \log(1 - y_{c_i})\right) \tag{11}$$

where $m$ is the total number of LSTM layer outputs ($h_m$) and $y_{t_i}$ and $y_{t_i}$ are the simulated and trans-lated adult-CM APs (the network output for translation task). The $y_{c_i}$ is binary indicator of class labels for iPSC-CM APs (0 for drug-free or 1 for drugged categories) and $y_{c_i}$ is predicted probability of APs being classified into the discussed classes. We used sum of both loss functions (*Equation 12*) to cal-culate the overall network error ($J$) for both translation and classification tasks during the network training process. We updated network parameters ($\theta_{t+1}$) using adaptive momentum estimation (ADAM) optimization algorithm (*Kingma and Ba, 2014*) based on the average gradient of overall loss function with respect to the network parameters for 64 randomly selected simulated AP traces (mini-batch = 64) at each training iteration (*Equations 13–15*).

$$J(\theta_t) = CrossEntropy_{Classification}(\theta_t) + \mathrm{MSE}_{Translation}(\theta_t) \tag{12}$$

$$\theta_{t+1} = \theta_t - \frac{\alpha.\hat{m}_t}{\sqrt{\hat{\nu}_t} + \epsilon}, \theta_t \epsilon \left\{ W_{i,j}^n, b_j^n \right\} \tag{13}$$

$$\hat{m}_t = \frac{m_t}{1 - \beta_1}, \mathrm{where}\, m_t = (1 - \beta_1)\nabla J(\theta_t) + \beta_1 m_{t-1} \tag{14}$$

$$\hat{\nu}_t = \frac{\nu_t}{1 - \beta_2}, \mathrm{where}\, \nu_t = (1 - \beta_2)(\nabla J(\theta_t))^2 + \beta_2 v_{t-1} \tag{15}$$

We used a rectified linear unit (ReLu) (*Glorot et al., 2011*) as activation function in *Equation 8* to calculate the output for each hidden layer neuron at each training iteration. We used dropout regu-larization (*Zaremba et al., 2014*) to randomly drop neurons with 0.2 probability of elimination along with their connections from the LSTM and fully connected layers during training to reduce the over-fitting. We kept updating the network parameters using ADAM optimization algorithm (*Equation 13*) to find global minimum of loss function (*Equation 12*). We computed the exponential average of the gradient (*Equation 14*) as well as the square of the gradient (*Equation 15*) for each parameter ($\theta_t$), where $\alpha$ is the learning rate equal to 0.001, $\beta_1, \beta_2$ are first and second momentum coefficients equal to 0.9 and 0.999, and $\epsilon$ is a small term equal to $1e^{-8}$ preventing division by 0.

## Computational workflow (*Figure 4*)

We first preprocessed iPSC-CM and adult-CM APs by applying a digital forward and backward data filtering technique (*Gustafsson, 1996*) and calculated the mean values for iPSC-CM and adult-CM AP traces. We removed the calculated mean values from the corresponding AP traces to center val-ues on zero. Next the iPSC-CM and adult-CM AP traces were divided by maximum AP values to nor-malize the AP values for more efficient training process. Next, we split the preprocessed data in 70:10:20 ratio into training, validation, and test datasets, respectively, and implemented the network architecture using Pytorch (*Ketkar, 2017*). During the training process, the multitask network received iPSC-CM AP time-course data as inputs and predicted adult-CM AP time courses. The net-work also received the category (drug-free and drugged) of the iPSC-CM AP data. The network next calculated the MSE (*Equation 10*) between predicted AP waveforms and the expected waveforms for adult-CM APs. It also calculated cross-entropy (*Equation 11*) between the predicted category for the iPSC-CM AP and the expected value. The cross-entropy was added to the calculated MSE to determine the total loss for training. The ADAM optimization algorithm was then used to update the network weights and bias terms.

We performed updating the network parameters (*Equation 13*) and monitored the network per-formance for the training and validation datasets until the point at which the network performance on the training dataset began to degrade compared to the validation dataset. This process was used to identify the optimal number of iterations (epochs = 300) for the training process. The last trained network was designated as the best possible model to perform both translation and classifi-cation tasks. We then used a holdout test dataset and calculated MSE (*Equation 10*), $R^2$ score (*Equations 16 and 17*), and the error in prediction for adult-CM $APD_{90}$ as evaluation metrics to assess the performance of the network for translation task and the AUROC, accuracy, recall, and pre-cision to measure capability of network for classification task as described below. The network codes

have been made publicly available at Clancy lab GitHub. (https://github.com/ClancyLabUCD/Multi-task_network).

## Evaluation metrics for the translation and classification tasks

As we discussed, we used MSE and cross-entropy loss functions for performance evaluation of translation and classification tasks. In addition to MSE, we computed $R^2$ score (*Devore, 2011*; *Equations 16 and 17*) to measure how close the translated adult-CM AP $\left(y_{t_i}\right)$ was to the expected simulated adult-CM AP $\left(y_{t_i}\right)$. We compared the histogram distribution of simulated and translated adult-CM $APD_{90}$ values and the error in $APD_{90}$ prediction to assess the accuracy of network prediction.

$$\bar{y}_{t_i} = \frac{1}{m}\sum_{i=1}^{m} y_{t_i} \tag{16}$$

$$R^2 = \frac{\sum_i \left(y_{t_i} - \bar{y}_{t_i}\right)}{\sum_i \left(y_{t_i} - \bar{y}_{t_i}\right)} \tag{17}$$

We used AUROC to measure the capability of the model to distinguish between drug-free and drugged iPSC-CM APs (*Fawcett, 2006*). AUROC is the area under the receiver operating characteristic (ROC) curve that is a plot of the false-positive rate (FPR), the probability that the network classified drug-free iPSC-CM APs into drugged categories (FP) (*Equation 18*) versus the true-positive rate (TPR) or recall, the probability that the network correctly classified drugged iPSC-CM APs into drugged category (TP) (*Equation 19*). AUROC close to 1 indicated a model with a desirable measure of separability, while a poor model had AUROC near 0, which means that it had poor separability.

In addition, we used recall, accuracy, and precision to describe the performance of the network for the classification task (*Sube and Ertel, 2017*), where the accuracy and precision indicated the proportion of all correct, TP + true negatives (TN), that is, predicted drug-free APs (*Equation 20*) and correct positive identifications (*Equation 21*). False negatives (FN) in *Equations 19 and 20* were the total number of drugged iPSC-CM APs classified as drug-free.

$$\mathrm{FPR} = \frac{FP}{FP + TN} \tag{18}$$

$$\mathrm{Recall} = \frac{TP}{TP + FN} \tag{19}$$

$$\mathrm{Accuracy} = 100 * \frac{TP + TN}{TP + TN + FP + FN} \tag{20}$$

$$Precision = \frac{TP}{TP + FP} \tag{21}$$

## Acknowledgements

This study was supported by the NIH Common Fund OT2OD026580, SPARC OT2OD025308-01S2, American Heart Association Career Development Award 19CDA34770101, NIH NHLBI grants R01HL152681, R01HL128170 and U01HL126273, Department of Physiology and Membrane Biology Research Partnership Fund, Extreme Science and Engineering Discovery Environment (XSEDE) Grant MCB170095, National Center for Supercomputing Applications (NCSA) Blue Waters Broadening Participation Allocation, Texas Advanced Computing Center (TACC) Leadership Resource Allocation MCB20010, Oracle cloud for research allocation.

## Additional information

### Funding

| Funder | Grant reference number | Author |
|---|---|---|
| NIH | OT2OD026580 | Igor Vorobyov<br>Colleen E Clancy |
| NIH | OT2OD025308-01S2 | Parya Aghasafari |
| American Heart Association | 19CDA34770101 | Igor Vorobyov |
| National Heart, Lung, and Blood Institute | R01HL152681 | Igor Vorobyov<br>Colleen E Clancy |
| National Heart, Lung, and Blood Institute | R01HL128170 | Colleen E Clancy |
| National Heart, Lung, and Blood Institute | U01HL126273 | Colleen E Clancy |
| UC Davis Department of Physiology and Membrane Biology | Research Partnership Fund | Igor Vorobyov<br>Colleen E Clancy |
| National Science Foundation | MCB170095 | Igor Vorobyov<br>Colleen E Clancy |
| National Centre for Supercomputing Applications | | Igor Vorobyov<br>Colleen E Clancy |
| Texas Advanced Computing Center | MCB20010 | Igor Vorobyov<br>Colleen E Clancy |
| Oracle | | Igor Vorobyov<br>Colleen E Clancy |

The funders had no role in study design, data collection and interpretation, or the decision to submit the work for publication.

### Author contributions

Parya Aghasafari, Conceptualization, Software, Formal analysis, Validation, Investigation, Visualization, Methodology, Writing - original draft, Writing - review and editing; Pei-Chi Yang, Conceptualization, Data curation, Software, Formal analysis, Validation, Investigation, Visualization, Methodology, Writing - original draft, Writing - review and editing; Divya C Kernik, Data curation, Methodology, Writing - original draft; Kazuho Sakamoto, Yasunari Kanda, Junko Kurokawa, Data curation, Validation; Igor Vorobyov, Investigation, Writing - review and editing; Colleen E Clancy, Conceptualization, Resources, Data curation, Formal analysis, Supervision, Funding acquisition, Investigation, Methodology, Writing - original draft, Project administration, Writing - review and editing

### Author ORCIDs

Yasunari Kanda https://orcid.org/0000-0003-2527-3526
Igor Vorobyov http://orcid.org/0000-0002-4767-5297
Colleen E Clancy https://orcid.org/0000-0001-6849-4885

### Decision letter and Author response

Decision letter https://doi.org/10.7554/eLife.68335.sa1
Author response https://doi.org/10.7554/eLife.68335.sa2

## Additional files

### Supplementary files
• Transparent reporting form

## Data availability

Since we used simulated data, we have made all drugged and drug-free iPSC-CM and adult-CM AP data used for training and testing the multitask network publicly available at Clancy lab Github. (https://github.com/ClancyLabUCD/Multitask_network/tree/master/data, copy archived at https://archive.softwareheritage.org/swh:1:rev:7f2b653a91f552d66ae2d9b70b720f8706b36da3). In addition, we have illustrated training and test dataset in Figure 1 and Figure 5. We have also shared the jupyter notebook for preparing clean and organized data for training the network at Clancy lab Github (https://github.com/ClancyLabUCD/Multitask_network/tree/master/jupyter). We also made experimental data used for the model validation publicly available at Clancy lab Github. (https://github.com/ClancyLabUCD/Multitask_network/blob/master/data/clean_data/experiments.csv ). Figure 7 illustrates the experimental data we used to validate the network.

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
