## [Decision Letter]

**Acceptance summary:**

The authors are to be commended on a rigorous, impactful and well-written manuscript describing an unbiasd computational tool for classifying and translating action potential waveforms from iPSC-CMs to those that would be expected in a mature adult CM. This work will likely be of interest to growing number of investigators using iPSC-CMs for drug screening and disease mechanisms.

**Decision letter after peer review:**

Thank you for submitting your work entitled "A deep learning algorithm to translate and classify cardiac electrophysiology: From induced pluripotent stem cell-derived cardiomyocytes to adult cardiac cells" for further consideration by *eLife*. Your article has been evaluated by two Reviewers, one of whom is a member of our Board of Reviewing Editors. This evaluation has been overseen by José Faraldo-Gómez as Senior Editor. As you will see below, there are several issues that would need to be resolved before your manuscript can be accepted for publication. We encourage you to address these issues as clearly and unequivocally as possible, and to submit a revised version of your manuscript.

*Reviewer #1:*

The authors set out to develop an unbiased computational tool for classifying and translating action potential waveforms from iPSC-CMs to those that would be expected in a mature adult CM. A deep-learning neural network approach is used for the model which avoids some of the limiting assumptions (for example, linearity) employed in previous modeling approaches. Two different established computational models are used to generate iPSC-CM and adult waveforms for training and testing the deep learning model. They show their approach is able to do a good job classifying waveforms into drug free and drugged and then translating an iPSC-CM AP into an adult AP. The model is further tested on experimentally recorded iPSC-CM APs and data in the literature. In general, the authors do a good job establishing the potential strengths of their approach, which will likely be of interest to growing number of investigators using iPSC-CMs for drug screening and disease mechanisms.

It would help from the onset to provide more clear rationale for the classification part of the model – it's easy to understand why one would want to translate from iPSC-CM to adult but how do the authors envision the classification piece being used? I could see this being useful potentially for diagnosis – but the rationale or potential application could be better spelled out.

A nice feature of the study is not just showing effective classification/translation but also doing some analysis to see what part of the waveform is most important for the translation (late repolarization phase). Can a similar analysis be done to determine what part is most important for the classification part? Do the authors expect the same part of the AP would be most important in determining drug vs drug free?

The validation of the drug block in figure 7 is a little confusing and could be more clear. Why were only 4 datapoints used? The range is narrow and comparison not that impressive (rate dependence is inverted at least for the data shown).

Some additional discussion about limitations of the approach and how the authors envision implementation in practice would be useful. iPSC-CMs properties can vary quite a bit depending on protocol for differentiation, etc. Presumably the network model developed here would only be useful for iPSCs generated in a specific manner. This should be more clearly discussed.

*Reviewer #2:*

Although iPSC-CMs have much potential as an in vitro model, their well-known limitations in mimicking adult cardiac electrophysiological behaviors is a serious shortcoming and puts limits of their utilization, e.g., for safety pharmacology. This manuscript provides an approach to overcome some of these shortcomings by translating the immature iPSC-CM action potentials into human adult-like action potentials. This is accomplished by a novel application of deep learning techniques. The learning algorithm is also capable of classifying between drug-free and drugged action potentials under conditions of IKr (a key ionic current in safety pharmacology) block.

Strengths:

The application of a deep learning network approach to classify and translate action potential will be of interest in the field of cardiac electrophysiology.

The main conclusions are well-supported by the data.

Weaknesses:

Previous work (including refs. 11, 12, and 20) have made headway into the problem of translating between iPSC-CM and adult electrophysiology using optimization or regression methods. Although these previous approaches rely on a number of assumptions (as mentioned by the authors), they do work quite well.

Limitations of the deep learning method are not discussed.

Comments for the authors:

The authors should specify how the normalization procedure was done and comment on the difference in baseline between iPSC-CM models (around -1) and adult modes (around -0.5) in figure 5.

The lack of longer APD values for the translated population in Figure 5B is not apparent in panel C. Similarly, in panel E the longer APDs would be expected to give a much more narrow second peak in panel F. The authors should clarify this.

The authors identify the 400-500 ms time frame as key to successful translation (Figure 6). The authors offer two explanations: an increase in membrane resistance (panel B) and a small total current (panel C). However, the latter explanation does not seem consistent with the figure: the total current is actually larger in that time frame than earlier in the plateau.

Only 4 action potential recordings are used to test the network on experimental data. More cells would generate a stronger validation.

The authors should discuss limitations of their approach.

---

## [Author Response]

Reviewer #1:The authors set out to develop an unbiased computational tool for classifying and translating action potential waveforms from iPSC-CMs to those that would be expected in a mature adult CM. A deep-learning neural network approach is used for the model which avoids some of the limiting assumptions (for example, linearity) employed in previous modeling approaches. Two different established computational models are used to generate iPSC-CM and adult waveforms for training and testing the deep learning model. They show their approach is able to do a good job classifying waveforms into drug free and drugged and then translating an iPSC-CM AP into an adult AP. The model is further tested on experimentally recorded iPSC-CM APs and data in the literature. In general, the authors do a good job establishing the potential strengths of their approach, which will likely be of interest to growing number of investigators using iPSC-CMs for drug screening and disease mechanisms.It would help from the onset to provide more clear rationale for the classification part of the model – it's easy to understand why one would want to translate from iPSC-CM to adult but how do the authors envision the classification piece being used? I could see this being useful potentially for diagnosis – but the rationale or potential application could be better spelled out.A nice feature of the study is not just showing effective classification/translation but also doing some analysis to see what part of the waveform is most important for the translation (late repolarization phase). Can a similar analysis be done to determine what part is most important for the classification part? Do the authors expect the same part of the AP would be most important in determining drug vs drug free?

Thank you for this important comment. We have conducted and included the results of an ablation study for the classification task. The 400-500 ms time frame again is shown to contain the most important information to perform the classification task. This is not surprising as the effect of I_Kr_ block is critical during the high resistance phase of the membrane potential. It is possible that other types of perturbations (i.e. Na channel blocker, ischemia) may lead to a different outcome in the ablation study and we are currently pursuing those studies. We have addressed this response in the Figure and legend related to Figure 6A and in the text in lines 270-293 and in the discussion.

The validation of the drug block in figure 7 is a little confusing and could be more clear. Why were only 4 datapoints used? The range is narrow and comparison not that impressive (rate dependence is inverted at least for the data shown).

Thank you for this important comment. We have conducted and included the results of an ablation study for the classification task. The 400-500 ms time frame again is shown to contain the most important information to perform the classification task. This is not surprising as the effect of I_Kr_ block is critical during the high resistance phase of the membrane potential. It is possible that other types of perturbations (i.e. Na channel blocker, ischemia) may lead to a different outcome in the ablation study and we are currently pursuing those studies. We have addressed this response in the Figure and legend related to Figure 6A and in the text in lines 270-293 and in the discussion.

Some additional discussion about limitations of the approach and how the authors envision implementation in practice would be useful. iPSC-CMs properties can vary quite a bit depending on protocol for differentiation, etc. Presumably the network model developed here would only be useful for iPSCs generated in a specific manner. This should be more clearly discussed.

The deep learning algorithm presented here has the benefit of automating feature extraction without any predetermination of the feature. It also allows for the translation of timecourse data from simulated or experimental data sets. However, there are some limitations to the approach. One limitation is the requirement for multiple datasets that are of sufficient quality for training – the more robust the training set, the higher the accuracy of the task. It is possible that this limitation can be addressed in future studies by developing new methods for data extraction and data interpolation in sparse data sets. We addressed this limitation by utilizing simulated data to train the network and this approach might be applicable for a variety of physiological problems. Simulated data can be generated to constitute a robust data set that can be used to train the multitask model and allow extraction of the relevant features from any timecourse data set. Please see the addition of this text in the second to last paragraph of the discussion.

The reviewer also provides an interesting new direction to pursue – might the classification task be able to distinguish cellular level signals derived from cells cultured using different protocols. We have added reference to this possibility in the last paragraph of the discussion.

Reviewer #2:Although iPSC-CMs have much potential as an in vitro model, their well-known limitations in mimicking adult cardiac electrophysiological behaviors is a serious shortcoming and puts limits of their utilization, e.g., for safety pharmacology. This manuscript provides an approach to overcome some of these shortcomings by translating the immature iPSC-CM action potentials into human adult-like action potentials. This is accomplished by a novel application of deep learning techniques. The learning algorithm is also capable of classifying between drug-free and drugged action potentials under conditions of IKr (a key ionic current in safety pharmacology) block.Strengths:The application of a deep learning network approach to classify and translate action potential will be of interest in the field of cardiac electrophysiology.The main conclusions are well-supported by the data.Weaknesses:Previous work (including refs. 11, 12, and 20) have made headway into the problem of translating between iPSC-CM and adult electrophysiology using optimization or regression methods. Although these previous approaches rely on a number of assumptions (as mentioned by the authors), they do work quite well.Limitations of the deep learning method are not discussed.

Thank you. The deep learning algorithm presented here has the benefit of automating feature extraction without any predetermination of the feature. It also allows for the translation of timecourse data from simulated or experimental data sets. However, there are some limitations to the approach. One limitation is the requirement for multiple datasets that are of sufficient quality for training – the more robust the training set, the higher the accuracy of the task. It is possible that this limitation can be addressed in future studies by developing new methods for data extraction and data interpolation in sparse data sets. We addressed this limitation by utilizing simulated data to train the network and this approach might be applicable for a variety of physiological problems. Simulated data can be generated to constitute a robust data set that can be used to train the multitask model and allow extraction of the relevant features from any timecourse data set. Please see the addition of this text in the second to last paragraph of the discussion.

Comments for the authors:The authors should specify how the normalization procedure was done and comment on the difference in baseline between iPSC-CM models (around -1) and adult modes (around -0.5) in figure 5.

First, the mean values for iPSC-CM and adult-CM AP traces were calculated and the calculated mean values were removed from corresponding AP traces to center values on zero. Next the maximum values of removed mean iPSC and adult-CM AP values were calculated and the removed mean values were divided by the calculated maximum values to map AP values between -1 and 1. We have addressed this in the methods section entitled “Computational Workflow” on Page 21.

The lack of longer APD values for the translated population in Figure 5B is not apparent in panel C. Similarly, in panel E the longer APDs would be expected to give a much more narrow second peak in panel F. The authors should clarify this.

As presented in figure 5C, F the second peak for translated AP traces (green) is taller and narrower compared to the simulated data (red) indicating higher frequency for APD_90_ values around 500 for translated adult-CM AP traces. The distributions in panels C and F tell the whole story – it’s not possible to “eyeball” the frequency of traces in panels B and E because there is so much overlap.

The authors identify the 400-500 ms time frame as key to successful translation (Figure 6). The authors offer two explanations: an increase in membrane resistance (panel B) and a small total current (panel C). However, the latter explanation does not seem consistent with the figure: the total current is actually larger in that time frame than earlier in the plateau.

Thank you for this feedback – we have clarified that the *change* in current (dI) id very close to zero when the current is flat during the 400-500 ms time period. The membrane resistance (Figure 6C red) as a function of simulation time indicating very high values (as dI 0, dV/dI ∞) for the latter at 400-500 ms.

Only 4 action potential recordings are used to test the network on experimental data. More cells would generate a stronger validation.

Thank you – we have added more datapoints (now 300 simulated iPSC-CMs translated and compared to data reported in the O’Hara-Rudy paper) to clarify the translator performance. At this time, we are unable to obtain more experimental data to a change in lab personnel, but utilized the four provided as a demonstration of the concept applied to experimental data. We have addressed this response in Figure 7C in the results and in the figure legend.

The authors should discuss limitations of their approach.

Thank you. The deep learning algorithm presented here has the benefit of automating feature extraction without any predetermination of the feature. It also allows for the translation of timecourse data from simulated or experimental data sets. However, there are some limitations to the approach. One limitation is the requirement for multiple datasets that are of sufficient quality for training – the more robust the training set, the higher the accuracy of the task. It is possible that this limitation can be addressed in future studies by developing new methods for data extraction and data interpolation in sparse data sets. We addressed this limitation by utilizing simulated data to train the network and this approach might be applicable for a variety of physiological problems. Simulated data can be generated to constitute a robust data set that can be used to train the multitask model and allow extraction of the relevant features from any timecourse data set. Please see the addition of this text in the second to last paragraph of the discussion.